

# Using bioinformatics and metabolomics to identify altered granulosa cells in patients with diminished ovarian reserve

Ruifen He[1], Zhongying Zhao[1], Yongxiu Yang[2] and Xiaolei Liang[2]

[1] The First Clinical Medical College of Lanzhou University, Lanzhou, China
[2] Department of Obstetrics and Gynecology, The First Hospital of Lanzhou University, Key Laboratory for Gynecologic Oncology Gansu Province, Lanzhou, China

## ABSTRACT

**Background**. During fertility treatment, diminished ovarian reserve (DOR) is a challenge that can seriously affect a patient's reproductive potential. However, the pathogenesis of DOR is still unclear and its treatment options are limited. This study aimed to explore DOR's molecular mechanisms.

**Methods**. We used R software to analyze the mRNA microarray dataset E-MTAB-391 downloaded from ArrayExpress, screen for differentially expressed genes (DEGs), and perform functional enrichment analyses. We also constructed the protein-protein interaction (PPI) and miRNA-mRNA networks. Ovarian granulosa cells (GCs) from women with DOR and the control group were collected to perform untargeted metabolomics analyses. Additionally, small molecule drugs were identified using the Connectivity Map database.

**Results**. We ultimately identified 138 DEGs. Our gene ontology (GO) analysis indicated that DEGs were mainly enriched in cytokine and steroid biosynthetic processes. According to the Kyoto Encyclopedia of Genes and Genomes (KEGG), the DEGs were mainly enriched in the AGE (advanced glycation end-product)-RAGE (receptor for AGE) signaling pathway in diabetic complications and steroid biosynthesis. In the PPI network, we determined that JUN, EGR1, HMGCR, ATF3, and SQLE were hub genes that may be involved in steroid biosynthesis and inflammation. miRNAs also played a role in DOR development by regulating target genes. We validated the differences in steroid metabolism across GCs using liquid chromatography-tandem mass spectrometry (LC-MS/MS). We selected 31 small molecules with potentially positive or negative influences on DOR development.

**Conclusion**. We found that steroidogenesis and inflammation played critical roles in DOR development, and our results provide promising insights for predicting and treating DOR.

## INTRODUCTION

Diminished ovarian reserve (DOR), defined as a decline in the number or quality of follicles and oocytes, reduces a female patient's reproductive potential (*Sharara, Scott Jr & Seifer,*

Corresponding author
Xiaolei Liang, liangxl07@lzu.edu.cn

*1998*). The incidence of DOR ranges from 9% to 24% in women undergoing in vitro fertilization (IVF) (*Kyrou et al., 2009*). The most common effective treatment for DOR is the use of assisted reproductive techniques. Ovarian reserve naturally declines with age, but some women experience DOR much earlier than average. Young women with DOR thus have an accelerated physiological decline in ovarian reserve. DOR is one of the greatest challenges for reproductive endocrinologists because it is characterized by poor ovarian response to gonadotrophin stimulation, low pregnancy rates, and high rates of pregnancy loss (*Levi et al., 2001*; *Navot, Rosenwaks & Margalioth, 1987*).

An ovarian follicle is a complex structure comprised of oocytes, cumulus cells (CCs), and granulosa cells (GCs). The bidirectional communication between the oocyte and its companion somatic cells is essential for follicular development and oocyte growth (*Anderson & Albertini, 1976*; *Matzuk et al., 2002*). In mice, the oocyte's transcriptional activity was modulated when in vitro cultured with GCs, but not without GCs (*De La Fuente & Eppig, 2001*). Another study showed that DOR patients had an increase in GC apoptosis, which is associated with a poor ovarian response and oocyte yield (*Fan et al., 2019*). Considering this coadjutant relationship between oocyte and GC, exploring the alteration of GCs from women with DOR may provide a deeper understanding for DOR pathogenesis. Previous studies have investigated the mRNA expression profiles of ovarian CCs (*Greenseid et al., 2011*) and GCs (*Chin et al., 2002*; *Skiadas et al., 2012*) in addition to miRNA expression patterns (*Chen et al., 2017*; *Woo et al., 2018*) in DOR patients. These studies typically had different inclusion criteria for DOR patients. miRNAs are small noncoding RNAs that play critical roles in many diseases and biological processes, including reproduction, and regulate mRNA translation and stability (*Fabian, Sonenberg & Filipowicz, 2010*). Exploring how miRNA affects female reproduction (*Sabry et al., 2019*) could reveal miRNA therapeutics as a possible DOR treatment option.

Steroid hormones are a type of steroid involved in many biological and physiological functions. Cholesterol is the precursor for steroid hormone synthesis (*Greaves et al., 2014*). Steroid hormone levels affect the follicular growth and development processes (*Chou & Chen, 2018*). Inflammation triggers a wide range of physiological and pathological processes (*Medzhitov, 2008*). Aberrant inflammation has a negative effect on folliculogenesis and ovulation, and polycystic ovary syndrome (PCOS) is associated with the chronic endogenous production of low-grade pro-inflammatory cytokines (*Boots & Jungheim, 2015*). Inflammation and steroidogenesis abnormalities may also be involved in the development of DOR.

In this study, we performed bioinformatic analyses on the mRNA expression profiles of DOR GCs from the publicly available dataset E-MTAB-391 in order to identify differentially expressed genes (DEGs). We constructed protein-protein interaction (PPI) networks based on the DEGs, and a miRNA-mRNA network using the differentially expressed miRNAs (DEMs) extracted from a previous study (*Woo et al., 2018*). Small molecule drugs with potential synergistic or antagonistic effects on DOR were also screened using the Connectivity Map (CMap) database (*Lamb et al., 2006*). Moreover, we applied liquid chromatography-tandem mass spectrometry (LC-MS/MS) on our samples to explore the
metabolic alteration of GCs. This study may shed light on the future of DOR prognosis and treatment.

# MATERIALS & METHODS

## Data collection

We screened GEO and ArrayExpress (https://www.ebi.ac.uk/arrayexpress/) for expressed profiles of GCs from patients with and without DOR. We excluded datasets that: (1) were without detailed sample information, (2) used samples based on cell lines or animal models, (3) had sample sizes <10, and (4) had quite different definitions of DOR or used different participant age ranges than our study. Only one mRNA microarray dataset, E-MTAB-391 (which included 13 DOR samples and 13 normal ovarian reserve (NOR) samples (*Skiadas et al., 2012*), met our criteria and was downloaded from ArrayExpress. E-MTAB-391′s platform was the A-MEXP-1564-IIIumina HumanRef-8 WG-DASL v3 Expression BeadChip. We directly extracted the data from the DOR miRNAs used in a previous study (*Woo et al., 2018*).

## Identifying DEGs

We used the limma package in R software (*Ritchie et al., 2015*) to identify DEGs across the DOR and NOR samples. The adjusted $P$-value (false discovery rate) was obtained using the Benjamini–Hochberg algorithm when screening the DEGs. We set $P < 0.05$ and |log 2 (fold change; FC) |>0.58 as the DEG cut-off criteria. The DEGs were divided into upregulated and downregulated DEGs and saved for subsequent analyses.

## Functional and enrichment analyses of DEGs

Gene ontology (GO) annotation is widely used to study the biological functions of multiple genes, and is comprised of three independent ontologies: biological process (BP), molecular function (MF), and cellular component (CC) (*Ashburner et al., 2000*). The Kyoto Encyclopedia of Genes and Genomes (KEGG) pathway analysis is a valuable tool used to assess the interaction networks of genes and their products (*Kanehisa et al., 2017*). In this study, we used the clusterProfiler package in R software to obtain the GO and KEGG pathway enrichment for the DEGs (*Yu et al., 2012*). We set the value of $P < 0.05$ as the threshold for significance.

## PPI network construction and module analysis

The STRING database offers PPI assessment and integration (*Szklarczyk et al., 2017*). To achieve a system-wide understanding across DEGs, we constructed the PPI network using STRING (version 10.5; https://string-db.org/cgi/input.pl) with a combined cutoff score $\geq 4$. The network was visualized using Cytoscape (version 3.6.1; https://cytoscape.org/), which can integrate biomolecular interaction networks into a unified conceptual framework (*Shannon et al., 2003*). The nodes and edges of the network represent proteins and protein-protein associations, respectively. We performed a module analysis of the PPI network based on the Molecular Complex Detection (MCODE) feature of the Cytoscape software using the following parameters: degree cut-off = 2, node score cut-off = 0.2, max depth =
100, and k-score = 2. We performed subsequent GO and KEGG pathway analyses of the selected modules using the clusterProfiler package.

## Exploring DEM target genes

We extracted 105 DEMs from the GCs of women diagnosed with DOR in a previous study (*Woo et al., 2018*). The multiMiR package integrated 11 miRNA-target databases (three validated and eight predicted miRNA-target databases) and three disease-/drug-related miRNA databases (*Ru et al., 2014*) to retrieve interactions between the DEMs and screened DEGs. The DEM target genes were only screened from the 11 miRNA-target databases. Finally, we visualized the regulatory miRNA-mRNA network using Cytoscape software.

## Identifying small molecules

The CMap database (https://portals.broadinstitute.org/cmap) contains 7,000 gene-expression profiles from cultured human cells treated with bioactive small molecules (*Lamb et al., 2006*), and is a valuable resource when looking for connections between diseases, genetic perturbation, and drug action. We mapped the upregulated and downregulated genes to the CMap database to identify potential small molecule drugs, which either have antagonistic or synergistic effects on DOR. We regarded n (the number of instances) $\geq 4$, enrichment >0.7, and $P$-value <0.01 as statistically significant.

## Collecting GCs

This study was approved by the ethics committee of the First Hospital of Lanzhou University (LDYYLL2019-44), and we obtained written informed consent from all participants. Ovarian GCs were collected from women with DOR ($n = 3$) and women with NOR ($n = 3$). All participants were $\leq 35$ years old to eliminate age as a potential confounding variable. DOR was identified using FSH levels ($12 \leq$ FSH <25) and ovarian response (the number of follicles on the day of the ovulatory human chorion gonadotropin (hCG) trigger injection $\leq 7$). We selected infertile women undergoing IVF due to male or tubal factor infertility for the NOR group. Controlled ovarian stimulation was performed and follicular development was monitored using a transvaginal ultrasound. Oocyte retrieval was performed 36 h after hCG administration and GCs were isolated from fluid aspirates using previously-described methods (*Vanacker et al., 2011*).

## LC-MS/MS experiments

We performed metabolite extraction and LC-MS/MS analysis at Beijing Genomics Institute (BGI). Each frozen GCs sample was thawed and weighed into 1.5 mL Eppendorf tubes. We added the internal standard solution and 800 μL of methanol/acetonitrile/water solvent (2:2:1, v/v/v) to homogenize. Mixtures were centrifuged at 25,000 rcf for 15 min, and the supernatant was transferred out and vacuum dried. We then re-extracted the metabolite extract in 200 μL of a methanol/water mixture (1:9, v/v). After vortexing, the samples were centrifuged again. We collected the supernatant and inspected each sample using 20 μL of supernatant, the Waters 2D UPLC (Waters, Milford, MA, USA), and a Q Exactive high-resolution mass spectrometer (Thermo Fisher Scientific, Waltham, MA, USA). The analytical column we used was an ACQUITY UPLC BEH C18 (1.7 μm, 2. 1×100 mm,

Waters). In positive ion mode, the mobile phase was MS-grade water with 0.1% formic acid (A) and 100% methanol with 0.1% formic acid (B). In negative ion mode, the mobile phase was MS-grade water with 10 mM of ammonium formate (A) and 95% methanol with 10 mM of ammonium formate (B). The extracts were gradient-eluted with a flow rate of 0.35 ml/min. The full scan and fragment acquisition resolutions were 70,000 and 17,500, respectively. The ESI parameters were set as follows: sheath gas flow rate of 40 L min-1, auxiliary gas flow rate of 10 L min-1, spray voltage of 3800 V (positive mode) and 3200 V (negative mode), capillary temperature of 320 °C, and auxiliary gas heater temperature of 350 °C. The LC-MS/MS data were processed using Compound Discoverer 3.0 software (Thermo Fisher Scientific). We identified the differential metabolites using a combination of principal component analysis (PCA) and univariate analysis.

## RESULTS

### Identifying DEGs related to DOR

In this study, we downloaded the normalized expression data from the E-MTAB-391 dataset (Fig. S1). A total of 18,128 genes were available for further DEG identification. Using our criteria, we selected 138 DEGs from the DOR and NOR samples, including 55 upregulated and 83 downregulated genes. The volcano plot in Fig. 1 shows the distribution of all screened genes. The heat map of all DEGs based on unsupervised hierarchical clustering is shown in Fig. S2.

### GO and KEGG pathway enrichment analyses of DEGs

According to the GO BP analysis, the upregulated DEGs were mainly enriched in skeletal muscle cell differentiation and regulation of transcription from RNA polymerase II promoter in response to stress. Downregulated DEGs were mainly enriched in the steroid biosynthetic and cholesterol biosynthetic processes. Figure 2 shows the top 20 GO BP up- and down-regulated DEG terms in detail. Additionally, we performed KEGG pathway enrichment analysis and the results can be found in Table 1. The upregulated DEGs were significantly enriched in the AGE (advanced glycation end-product)-RAGE (receptor for AGE) signaling pathway in diabetic complications and human T-cell leukemia virus 1 infection, and the downregulated DEGs were mainly enriched in steroid biosynthesis (Fig. 3) and terpenoid backbone biosynthesis (Fig. S3).

### Constructing PPI networks and module analysis

We imported all DEGs related to DOR into the STRING database when constructing the PPI network, which included 99 nodes and 294 edges when the cut-off combined score was set at ≥ 0.7. The PPI network was visualized using Cytoscape software (Fig. 4A). The genes with higher node degrees were Jun proto-oncogene, AP-1 transcription factor subunit (JUN, degree=24); early growth response 1 (EGR1, degree=18); 3-hydroxy-3-methylglutaryl-CoA reductase (HMGCR, degree=17); activating transcription factor 3 (ATF3, degree=15); and squalene epoxidase (SQLE, degree=15). Additionally, we filtered out the top two modules from the PPI network in order to implement further GO and KEGG pathway analyses. Module 1 (Fig. 4B) contained 12 downregulated genes enriched

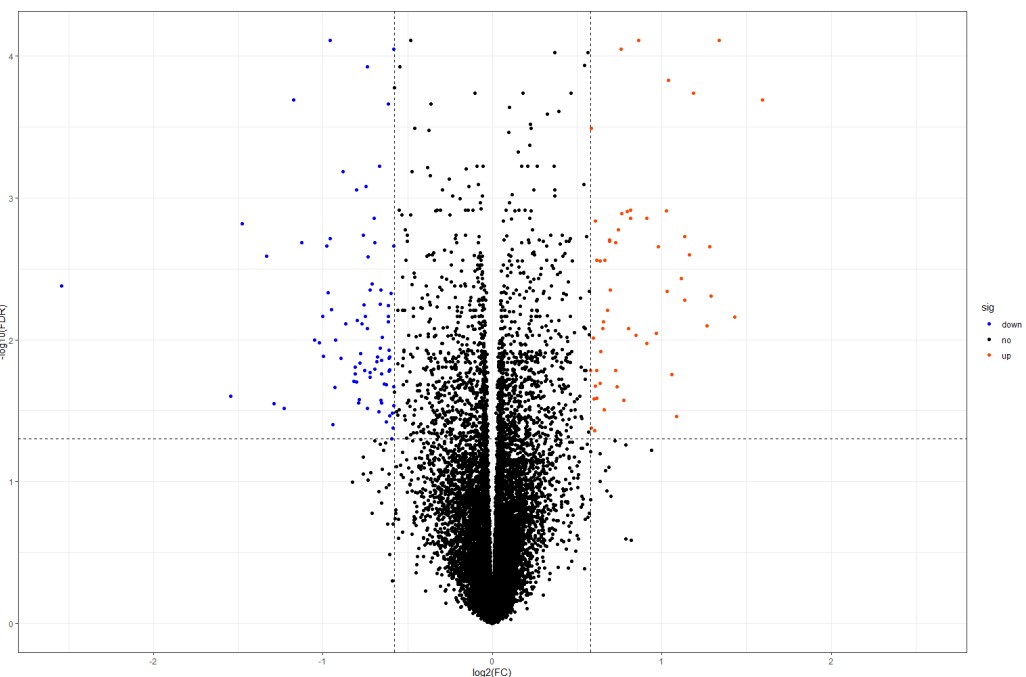

**Figure 1 Volcano plot of all DEGs.** The orangered and blue dots represent significantly upregulated and downregulated DEGs, respectively. DEGs, differentially expressed genes; FC, fold change.

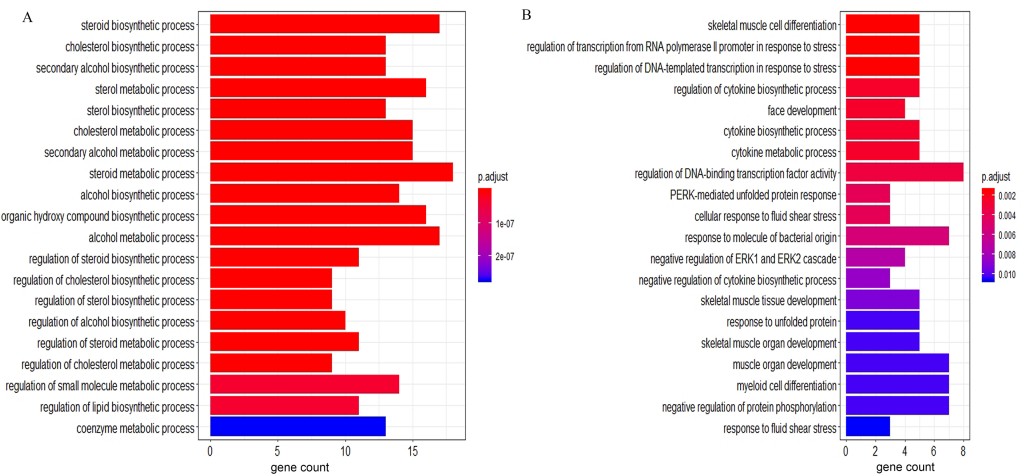

**Figure 2 The GO enrichment analysis of the DEGs.** (A) Top 20 enriched BP terms of downregulated genes. (B) Top 20 enriched BP terms of upregulated genes. The length of bars represents the number of genes, the color of bars represents corresponding adjusted *P*-value. GO, Gene Ontology; DEGs, differentially expressed genes; BP, biological process.

**Table 1** Top five KEGG pathways of the DEGs.

| Pathway ID | Description | P-value | Count | Genes | State |
|---|---|---|---|---|---|
| hsa04933 | AGE-RAGE signaling pathway in diabetic complications | 0.000389 | 4 | SERPINE1, EGR1, CXCL8, JUN | upregulated |
| hsa05166 | Human T-cell leukemia virus 1 infection | 0.000905 | 5 | EGR2, ZFP36, SERPINE1, SRF, JUN | upregulated |
| hsa05161 | Hepatitis B | 0.002359 | 4 | EGR3, EGR2, CXCL8, JUN | upregulated |
| hsa04657 | IL-17 signaling pathway | 0.004284 | 3 | FOSB, CXCL8, JUN | upregulated |
| hsa05203 | Viral carcinogenesis | 0.005122 | 4 | EGR3, EGR2, SRF, JUN | upregulated |
| hsa00100 | Steroid biosynthesis | 2.49E−11 | 7 | CYP51A1, FDFT1, MSMO1, EBP, NSDHL, SC5D, SQLE | downregulated |
| hsa00900 | Terpenoid backbone biosynthesis | 1.39E−05 | 4 | HMGCS1, HMGCR, IDI, MVD | downregulated |
| hsa04913 | Ovarian steroidogenesis | 0.00035 | 4 | LHCGR, STAR, HSD3B2,CYP19A1 | downregulated |
| hsa04066 | HIF-1 signaling pathway | 0.000908 | 5 | HK2, PGK1, ALDOC, ENO2, SLC2A1 | downregulated |
| hsa00010 | Glycolysis / Gluconeogenesis | 0.001222 | 4 | HK2, PGK1, ALDOC, ENO2, | downregulated |

**Notes.**

KEGG, Kyoto Encyclopedia of Genes and Genomes; DEGs, differentially expressed genes.

in the cholesterol biosynthetic process of the GO BP term. Module 2 (Fig. 4C) contained 10 upregulated genes that were mainly enriched in skeletal muscle cell differentiation of the GO BP term. The detailed results of the modules' GO and KEGG pathway analyses are shown in Table 2.

## Predicting DEM target genes and constructing the DEM-DEG regulatory network

We analyzed the DEMs associated with DOR, including 85 upregulated and 20 downregulated genes, using the multiMiR package to predict their target genes. We then used the identified DEM-DEG pairs (comprised of 91 DEMs and 109 DEGs) to construct the regulatory network. Among the pairs, miR-155-5p, miR-16-5p, let-7b-5p, miR-107, and miR-103a-3p had the most target genes. The detailed interactions between the DEMs and DEGs are shown in Fig. 5.

## Screening small molecule drugs

In order to screen out small molecule drugs, we compared all DEGs to the gene expression profiles in CMap. We identified 31 small molecules, seven of which had negative scores with the potential to reverse DOR. The detailed results are shown in Fig. 6.

## Metabolic differences between DOR and NOR GCs

Metabolites are essential for cellular function and untargeted metabolomics analyses can provide information on their associations with diseases. We analyzed the GC samples from the DOR and NOR groups using LC-MS/MS in both positive and negative ion modes. After data processing and metabolite identification, we screened the differential metabolites using a threshold $p$-value $<0.05$ and FC $\geq 1.2$ or $\leq 0.83$. We did detect metabolic differences between the GCs of the DOR and NOR samples. The detailed differences in the steroids and metabolites observed in the GCs of the two groups are listed in Table 3.

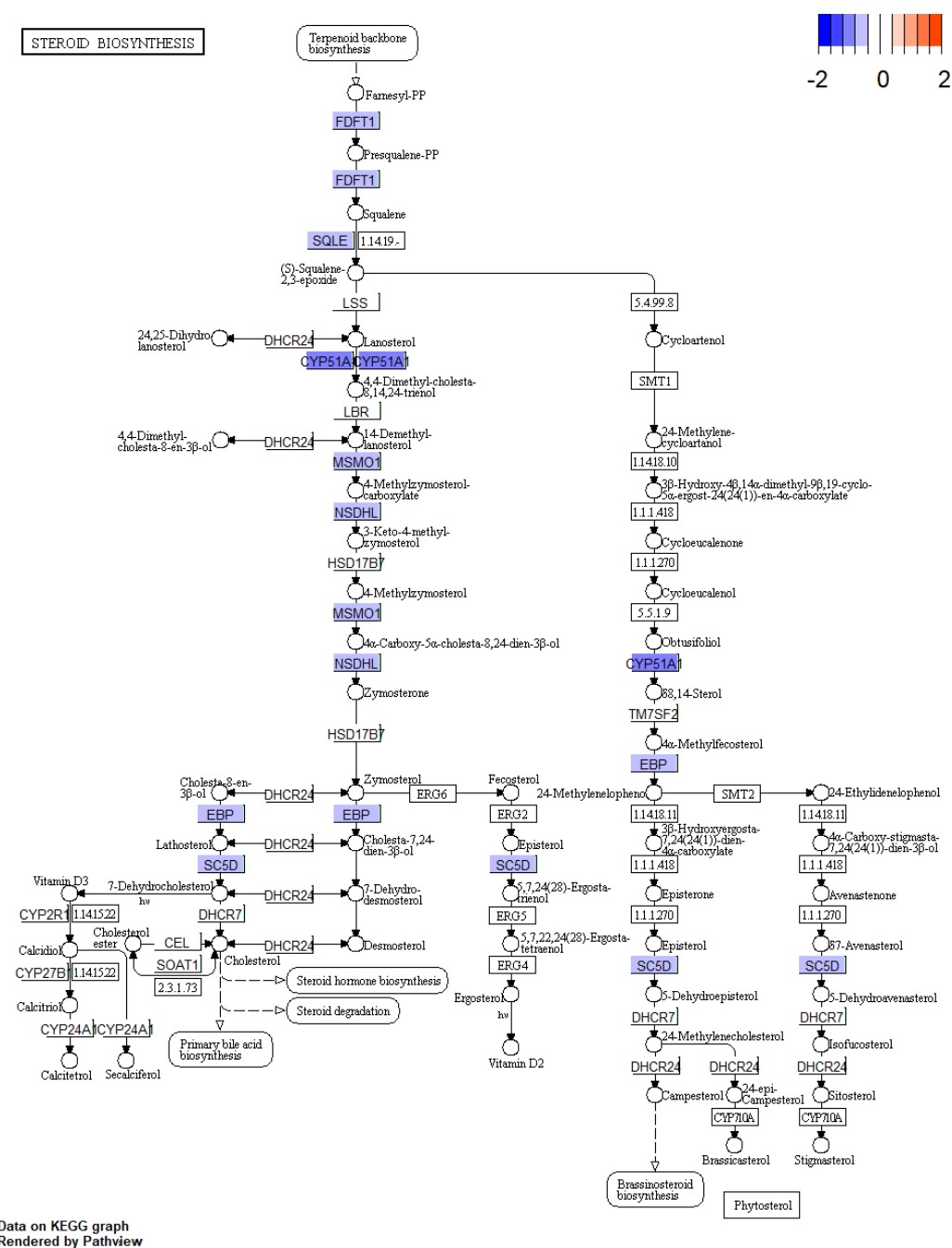

**Figure 3  Pathway of steroid biosynthesis from KEGG.** The genes with blue are downregulated DEGs. KEGG, Kyoto Encyclopedia of Genes and Genomes; DEGs, differentially expressed genes.

## DISCUSSION

A patient with DOR has a reduced number of retrieved oocytes compared to other women of a similar age. In some women, DOR can progress to a diagnosis of primary ovarian insufficiency (POI), which is an extreme form of ovarian dysfunction (*Cooper et al., 2011*).
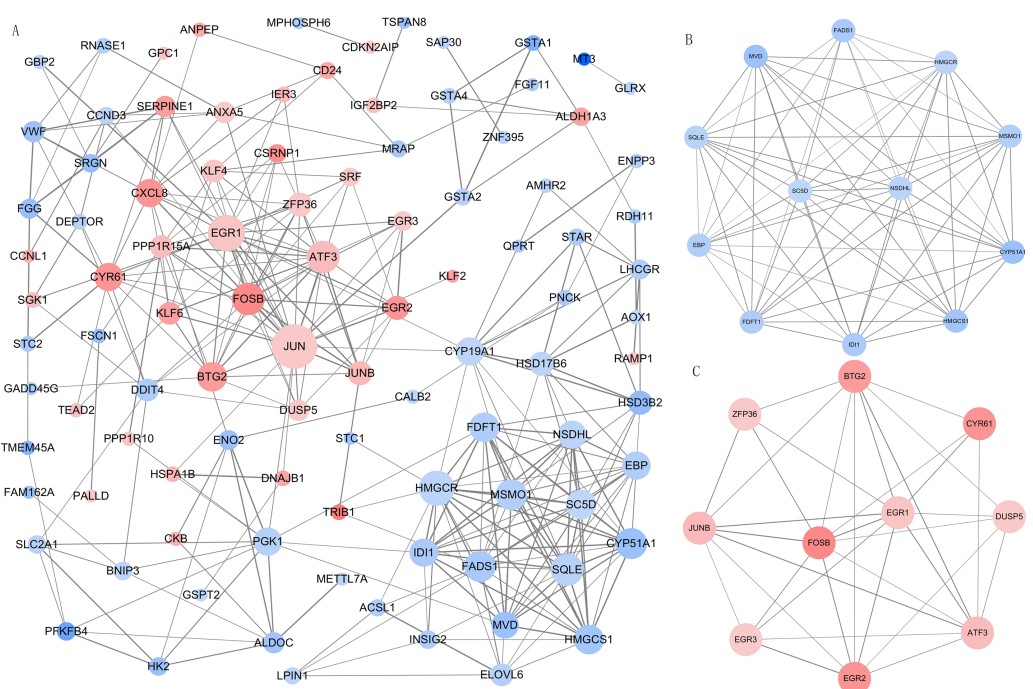

**Figure 4** **The PPI network and modular analysis.** (A) The PPI network of DEGs. (B) Module 1. (C) Module 2. Node represents gene, node size represents node degree. The orangered and blue nodes represent upregulated and downregulated genes, respectively. The depth of color represents the level of differential expression and the width of lines represents the combined score between two proteins. PPI, protein-protein interaction; DEGs, differentially expressed genes.

Because of this serious threat to a patient's reproductive health, there is an urgent need to further examine DOR etiology. Previous studies have used the GC mRNA/miRNA expression profiles from DOR patients to explore the molecular mechanisms of DOR. In this study, we focused on altered GCs from young women with DOR, and performed LC-MS/MS experiments and bioinformatic analyses to explore the differences between DOR and NOR. We obtained raw mRNA expression patterns and DEM data from previous publications (*Skiadas et al., 2012*; *Woo et al., 2018*) with similar inclusion criteria to ours. The inclusion criteria are presented in Table S1.

Sex steroid hormones (progestogens, androgens, and estrogens) have a steroid nucleus structure and are typically synthesized from cholesterol in the gonads and adrenal glands (*Greaves et al., 2014*). These hormones play important roles in female reproduction. The synthesis and secretion of estrogen are promoted by the elevated FSH levels found in patients with DOR (*Practice Committee of the American Society for Reproductive Medicine, 2015*). However, whether there is a difference in the estrogen levels of patients with DOR and NOR remains controversial. In our study, we found that downregulated DEGs were mainly enriched in the steroid biosynthetic process of the GO BP term (Fig. 2A). KEGG pathway analysis (Table 1) showed that downregulated DEGs were mainly enriched in steroid biosynthesis and terpenoid backbone biosynthesis. Therefore, a range of steroidogenesis substances may play a major role in DOR development. Consistent with our

**Table 2  Enriched GO BP terms (top five) and significantly enriched KEGG pathways of genes in the top two modules.**

| Modules | | Description | P.adjust | Count |
|---|---|---|---|---|
| module 1 | BP terms | cholesterol biosynthetic process | 3.53E−24 | 11 |
| | | secondary alcohol biosynthetic process | 3.53E−24 | 11 |
| | | sterol biosynthetic process | 4.99E−24 | 11 |
| | | cholesterol metabolic process | 2.88E−21 | 11 |
| | | secondary alcohol metabolic process | 3.39E−21 | 11 |
| | KEGG pathway | Steroid biosynthesis | 1.02E−15 | 7 |
| | | Terpenoid backbone biosynthesis | 1.09E−07 | 4 |
| module 2 | BP terms | skeletal muscle cell differentiation | 2.30E−05 | 4 |
| | | muscle organ development | 0.00019411 | 5 |
| | | skeletal muscle tissue development | 0.00019411 | 4 |
| | | skeletal muscle organ development | 0.00019411 | 4 |
| | | regulation of nuclear-transcribed mRNA poly(A) tail shortening | 0.001616513 | 2 |
| | KEGG pathway | Human T-cell leukemia virus 1 infection | 0.019978459 | 3 |
| | | C-type lectin receptor signaling pathway | 0.043059356 | 2 |
| | | Osteoclast differentiation | 0.043059356 | 2 |

**Notes.**
GO, Gene Ontology; BP, biological process; KEGG, Kyoto Encyclopedia of Genes and Genomes.

PPI network results (Fig. 4), we found several key genes in the top 20 (including HMGCR, SQLE, CYP51A, HMGCS1, FDFTI, SC5D, NSDHL, IDI1, EBP, and MSMO1) in steroid biosynthesis and terpenoid backbone biosynthesis pathways (Fig. 3 and Fig. S3). All these genes were downregulated in patients with DOR from our dataset. Previous studies have found that HMGCR catalyzes the first rate-limiting step in cholesterol synthesis (*Howe et al., 2017*); HMGCS1 condenses acetyl-CoA to form 3-hydroxy-3-methylglutaryl CoA, which is the substrate for HMGCR (*Mathews et al., 2014*); and CYP51A also participates in cholesterol synthesis, which can catalyze the removal of the 14α-methyl group from lanosterol (*Sharpe & Brown, 2013*). Upstream biological disruptions lead to a series of metabolomic changes. According to our untargeted metabolomics analysis, several steroids (Table 3) were significantly lower in GCs from patients with DOR compared to the control group. Among these steroids, progesterone plays an essential role in female reproductive events (ovulation, implantation, and pregnancy maintenance) and serves as an intermediate during estrogen biosynthesis (*Gellersen, Fernandes & Brosens, 2009*). Prior evidence suggests that GCs can directly produce progesterone before entering theca cells to convert into androgens (*Oktem et al., 2017*). Hydroxyprogesterone acts as an intermediate during the conversion of progesterone to androgens that are transported into GCs and converted into estrogen. The relationships between steroids and ovarian function (Table 3) have been rarely reported on and require more study. It has been demonstrated that several steroidogenic gene disturbances induced by bisphenol A can cause developmental impairments of ovary tissue (*Liu et al., 2019*). Overall, patients with DOR may have impaired hormone synthesis, which could be compensated for by elevated FSH levels. The perturbation of steroidogenic genes may be responsible for DOR development.

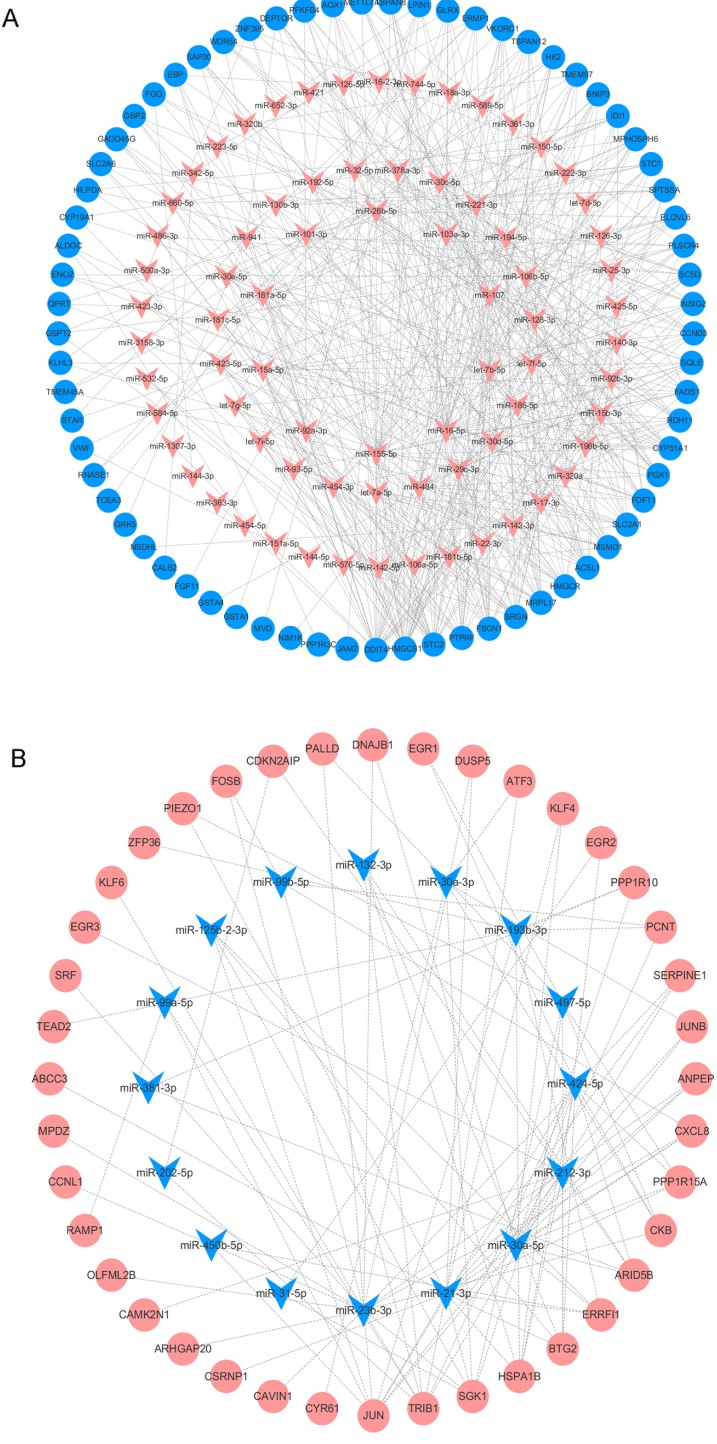

**Figure 5   The DEMs-DEGs regulatory network.** (A) The upregulated DEMs and targeted DEGs. (B) The downregulated DEMs and targeted DEGs. Circle nodes represent DEGs and triangle nodes represent DEMs. Orangered represent upregulation and blue represent downregulation. DEGs, differentially expressed genes; DEMs, differentially expressed miRNAs.

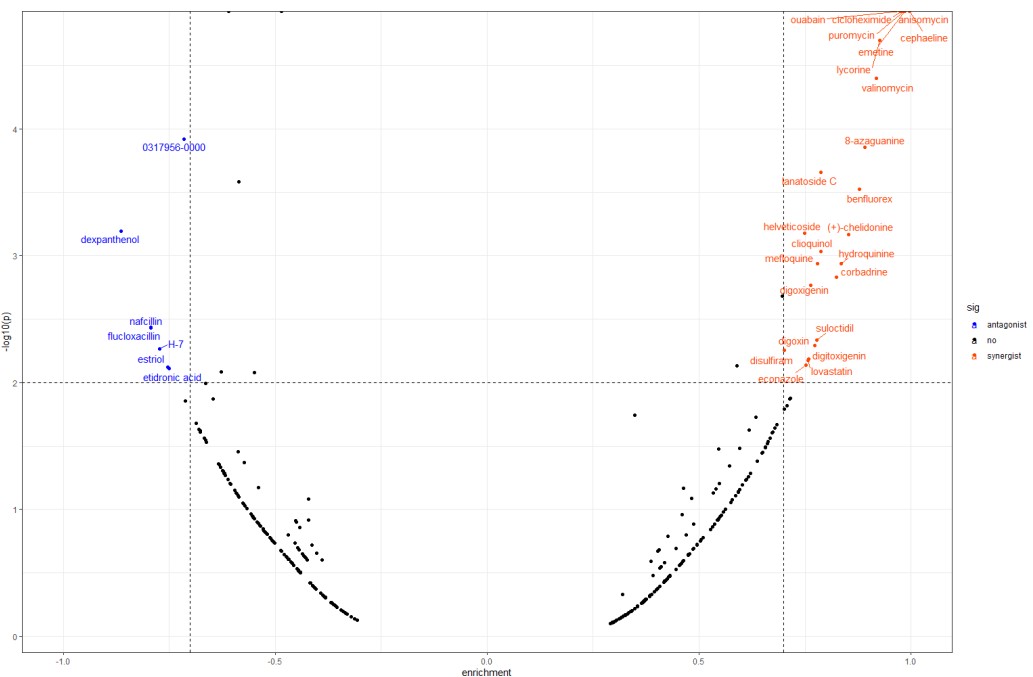

**Figure 6 Result of CMap analysis.** The orangered dots represent the synergistic small molecule drugs. The blue dots represent antagonistic small molecule drugs. CMap, Connectivity Map.

Aberrant inflammation has been suggested to influence follicular growth and development (*Boots & Jungheim, 2015*). Our results showed that upregulated genes were enriched in the AGE-RAGE signaling pathway (Table 1). The AGE-RAGE signaling pathway has been shown to induce reactive oxygen species (ROS) burst and inflammation, eventually leading to POI (*Huang et al., 2019*). EGR1 plays a proinflammatory role in numerous pathological processes and human diseases (*Schmidt et al., 2008*). A recent study found that in mice, EGR1 was increased in aged ovaries compared to young ovaries (*Yuan et al., 2016*). Our results showed that EGR1 as a key gene in the PPI network was upregulated in women with DOR. During cholestasis, EGR1 regulates the production of inflammatory mediators, including cytokines and adhesion molecules, that promote the accumulation and activation of inflammatory cells, causing liver injuries (*Bonetti et al., 2010*). According to the results of our GO analysis of upregulated DEGs (Fig. 2B), cytokines may be associated with DOR development. Cytokines play a key role in inflammation, and can be found in the immune cells of the ovary (*Tabibzadeh, 1994*; *Vinatier et al., 1995*). Accumulated evidence suggests that inflammation is closely related to ovarian dysfunction. Women diagnosed with PCOS often present with chronic low-grade inflammation due to overactive interleukin-1 (IL-1), a proinflammatory cytokine (*Popovic, Sartorius & Christ-Crain, 2019*). Additionally, multiple autoimmune diseases have adverse effects on female fertility via premature DOR (*Sen et al., 2014*). Therefore, anti-inflammatory treatment may be able to alleviate the progression of DOR. In a POI rat model, resveratrol counteracted inflammatory signaling induced by ionizing radiation, and preserved the entire ovarian follicle pool (*Said et al.,*

*2016*). More studies are needed to confirm the role of inflammation in DOR development and whether controlling inflammation is an option for DOR treatment.

miRNAs, a class of endogenous non-coding small molecule RNA, play an important role in gene expression modulation at the post-transcriptional level. Previous studies have shown that miRNAs help regulate reproductive functions, particularly follicular development, oocyte maturation, corpus function, pregnancy establishment, and early embryonic development (*Eisenberg et al., 2015*; *Tesfaye et al., 2016*). The role of miRNAs in ovarian function has been demonstrated primarily by the conditional knockout of Dicer (*Luense, Carletti & Christenson, 2009*), a cytoplasmic RNase; required for miRNA production in mammals. In a mouse model, the conditional knockout of Dicer in ovarian GCs led to decreased ovulation rates (*Nagaraja et al., 2008*) and compromised folliculogenesis and POI in oocytes (*Yuan et al., 2014*). In our study, we found that steroidogenic genes were regulated by differentially expressed miRNAs. HMGCS1 was regulated by 25 DEMs, including miR-155-5p, miR-16-5p, let-7b-5p, miR-107, and miR-103a-3p. Furthermore, a single miRNA can target multiple genes. MiR-107 targets 21 DEGs, including five steroidogenic genes: HMGCS1, FDFT1, CYP51A1, SQLE, and EBP. miRNA-107 expression in murine ovarian GCs exposed to cadmium was significantly different from expression in the control group, and miRNA-107 can regulate kit ligand (kitl) expression (*Wang et al., 2018*). Kitl plays an important role in the recruitment of primitive follicles (*Parrott & Skinner, 1999*), the proliferation and differentiation of GCs, the recruitment of theca cells, and early steroid hormone synthesis (*Flanagan, Chan & Leder, 1991*). Therefore, miRNAs may contribute to DOR development by regulating target genes. microRNA therapies for several diseases have reached clinical testing stages with promising results (*Rupaimoole & Slack, 2017*). miRNAs should be researched for potential DOR treatments.

We also conducted CMap analysis to quickly identify molecule drugs with antagonistic or synergistic effects on DOR based on their gene expression profiles. We found seven agents with negative scores that had potential for DOR treatment (Fig. 6). Among these, H-7 (1-(5-isoquinolinesulfonyl)-2-methylpiperazine), an inhibitor of protein kinase C, has been found to reduce the release of oocytes from rat ovaries (*Shimamoto, Yamoto & Nakano, 1993*). Estriol is a form of estrogen, and a meta-analysis concluded that luteal estradiol stimulation in assisted reproductive technology could decrease cycle cancellation rates and increase clinical pregnancy rates in poor responders exposed to controlled ovarian hyperstimulation (*Reynolds et al., 2013*). Therefore, we hypothesized that these molecule drugs, identified by bioinformatic analysis, may provide novel DOR treatment. Further validation of their effects is still needed.

Despite our study's promising findings, there were still limitations. We identified possible DEGs using |log2FC|>0.58 (the approximate fold change was >1.5), which is a relatively lower criterion than those used by other studies. DOR is an early stage ovarian reserve impairment that may take several years to develop into POI, and subtle alterations may have broader significance during its development. Small gene expression changes are also worth noting. In addition, we derived the E-MTAB-391 data from a large sample size. Nevertheless, we validated the steroid metabolism differences between DOR and NOR

He et al. (2020), *PeerJ*, DOI 10.7717/peerj.9812

**Table 3**  LC-MS/MS detected steroids that varied in GCs of DOR with significant difference.

| Components | Formula | m/z | FC | *p*-value | Class | Sub class | label |
|---|---|---|---|---|---|---|---|
| Hydroxyprogesterone | $C_{21}H_{30}O_3$ | 330.2190 | 0.0404 | 0.0381 | Steroids and steroid derivatives | Pregnane steroids | down |
| Progesterone | $C_{21}H_{30}O_2$ | 314.2244 | 0.0785 | 0.0025 | Steroids and steroid derivatives | Pregnane steroids | down |
| 3 alpha-hydroxydesogestrel | $C_{22}H_{30}O_2$ | 326.2241 | 0.0657 | 0.0027 | Steroids and steroid derivatives | Estrane steroids | down |
| (6beta,8xi,11beta,14xi,16alpha)-9-fluoro-6,11,17,21-tetrahydroxy-16-methylpregna-1,4-diene-3,20-dione | $C_{22}H_{29}FO_6$ | 408.1960 | 0.0290 | 0.0093 | Steroids and steroid derivatives | Hydroxysteroids | down |
| (6beta,8xi,11beta,14xi,16alpha)-9-fluoro-6,11,17,21-tetrahydroxy-16-methylpregna-1,4-diene-3,20-dione | $C_{22}H_{29}FO_6$ | 408.1959 | 0.0340 | 0.0179 | Steroids and steroid derivatives | Hydroxysteroids | down |
| 4,6-cholestadien-3-one | $C_{27}H_{42}O$ | 382.3229 | 0.0343 | 0.0352 | Steroids and steroid derivatives | Cholestane steroids | down |
| Cholest-4-en-3-one | $C_{27}H_{44}O$ | 384.3385 | 0.0796 | 0.0273 | Steroids and steroid derivatives | Cholestane steroids | down |

**Notes.**

LC-MS/MS, liquid chromatography-tandem mass spectrometry; GCs, granulosa cells; DOR, diminished ovarian reserve; FC, fold change.

samples using LC-MS/MS. Therefore, our findings are reliable and provide valuable insight into DOR development.

## CONCLUSION

We used bioinformatics approaches to investigate the perturbed steroidogenic and inflammation-related genes that may be regulated by miRNAs in women with DOR. Using metabolomics, we found that steroid metabolites were reduced in the GCs from DOR samples. Additionally, several small molecule drugs (e.g., the steroid hormone estriol) with potential antagonistic or synergistic effects on DOR were screened out. Our results suggest that steroidogenesis and inflammation play critical roles in DOR development, and should be pursued in future studies on DOR prediction and treatment.

### Funding

This study was supported by the National Natural Science Foundation of China (no. 81960278 and 81601351), and the Natural Science Foundation of the First Hospital of Lanzhou University (no. ldyyyn2018-52). The funders had no role in study design, data collection and analysis, decision to publish, or preparation of the manuscript.

### Grant Disclosures

The following grant information was disclosed by the authors:

The National Natural Science Foundation of China: 81960278, 81601351.

The Natural Science Foundation of the First Hospital of Lanzhou University: ldyyyn2018-52.

### Competing Interests

The authors declare there are no competing interests.

### Author Contributions

- Ruifen He conceived and designed the experiments, performed the experiments, authored or reviewed drafts of the paper, and approved the final draft.
- Zhongying Zhao performed the experiments, analyzed the data, prepared figures and/or tables, and approved the final draft.
- Yongxiu Yang analyzed the data, prepared figures and/or tables, and approved the final draft.
- Xiaolei Liang conceived and designed the experiments, authored or reviewed drafts of the paper, and approved the final draft.

### Human Ethics

The following information was supplied relating to ethical approvals (i.e., approving body and any reference numbers):

This study was approved by the Ethics Committee of The First Hospital of Lanzhou University (LDYYLL2019-44).

## Data Availability

The raw data are available in the Supplemental File.

## Supplemental Information

Supplemental information for this article can be found online at http://dx.doi.org/10.7717/peerj.9812#supplemental-information.

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
