# Peer review of "Using bioinformatics and metabolomics to identify altered granulosa cells in patients with diminished ovarian reserve"

_PeerJ, doi:10.7717/peerj.9812_

## Round 0.1 · original submission · Major Revisions

Several major issues have been raised by reviewers. The authors need to provide rationale/justification for the selection of parameters and the dataset. In addition, the authors should make metabolomics data fully available.

Reviewer 1 ·

Basic reporting

The authors conducted this study aiming to understand the molecular mechanisms behind diminished ovarian reserve with bioinformatic approaches. They utilized published gene expression data, including mRNA and miRNA dataset, and further analyzed and integrated them with several bioinformatic tools. They found inflammatory and steroidogenic signalings could responsible for diminished ovarian reserve. And they also supplemented this observation with their own metabolomic experiment which showed decreased steroids in the granulosa cells from patients with diminished ovarian reserve. The manuscript is well-written and logical sound. Several concerns will be listed below.

Experimental design

1. Most of the analysis in this manuscript highly depends on the E-MTAB-391 dataset the authors retrieved from a previous publication (Skiadas CC, et al.). However, there are multiple studies compared the gene expression of granulosa cells between normal and diminished ovarian reserve individuals (Greenseid K, et al., Chin K-V, et al., Pashaiasl M, et al.). These studies should be cited in the manuscript and the authors should justify the reason they only select a specific dataset as their input. For example, couple of the studies observed the changes in expression of IGF1 and IGF2. However, they are not on the DEG list of this manuscript. Why is that? And then why is the current dataset more optimal for the authors' purpose?

2. To identify differential expressed genes, the authors use log2 (FC) >0.58 as their cutoff. Several studies used log2 (FC) >1 as filter and it is not necessary the standard way to do it. But the authors has the responsibility to explain how this number (0.58) was chosen. If the filter was log2 (FC) >1, there would be no genes related to steroidogenesis in the list and the steroidogenesis is one of the major players being discussed in this manuscript.

3. For the differential expressed miRNA, the authors directly used the list from the publication of Chen D., et al. Again, it is not the only study toward this question. For example, Woo I., et al. did deep sequencing on the granulosa cells from normal and diminished ovarian reserve individuals. The differential expressed miRNA in their study are not the same list as in the study of Chen D., et al. So the authors should justify why they chose one over the other one (and please cite the publication) or they should consider to use both of them as input.

4. The authors then utilized the differential expressed genes and miRNAs to study the miRNA-mRNA regulatory network. The intention here is good. However, the authors did not compare the patient selection and sample collection methods in details. The authors should make a table to compare different cohorts of patients in their manuscript which should include their own cohort for the study of metabolomics. And the authors also need to explain how the differences between different selection methods could influence the results of their own if there is any. For example, the FSH level of these three cohorts are different.

Validity of the findings

1. In general, I appreciate the authors presented their data pretty well as well as discussed their findings in details in the discussion. At the same time, I do think the figures can be upgraded a little. The major concern here is that most of them will not be readable (figure2-6). There are a lot of labels and the fonts are way too small. In Figure 3, as far as I can see, there is only one color. And it is very hard to tell if the darkness of the green are different from each other. Please try to present the figures in a way the readers can read and digest.

2. I believe the metobolomic study in the end of paper can be interesting and beneficial to the filed. But the authors should provide the whole list (now there is only metabolites related to steroidogenesis) of their discovery as a supplementary file.

3. Please also provide the list of differential expressed miRNA used in this publication.

Additional comments

I believe the study can provide valuable information to the field. But as a manuscript heavily replies on bioinformatics, the selection of input is very critical. The authors need to be very careful on the way they select dataset and provide strong rationale on how they select the data.

Reviewer 2 ·

Basic reporting

no comment

Experimental design

no comment

Validity of the findings

no comment

Additional comments

Comments for the Author:

The manuscript by He et al was designed to understand the molecular mechanism of diminished ovarian reserve. They provide some preliminary data on the effects of steroidogenesis and inflammation at diminished ovarian reserve. Although observations made here are of interest, this study is rather too descriptive, and lack of mechanistic insights. Also, the writing is not very competent, with many awkward sentences.

Additional minor comments:

1: The authors need to proof read the entire manuscript carefully, and to make correct statements.

2: In results part, these titles are not interpret fitly.

---

## Round 0.2 · Minor Revisions

The authors appropriately addressed issues raised by reviewers.
Nevertheless, as one of reviewers suggests, the authors may want to extend the section that describes the role of miRNAs in the ovary, including some functional studies. Such change would improve the readability significantly.

Reviewer 1 ·

Basic reporting

See below

Experimental design

See below

Validity of the findings

See below

Additional comments

I appreciate the responses from the authors to each comments. The current manuscript is better scientifically and logically sound. However, the writing is somehow downgraded in this version, particularly for the newly added context. It is not only that the structure of the sentences is not reader-friendly but the contents are not professionally written. For example, the authors added a paragraph to describe the importance of miRNA in reproductive system. The most compelling evidence is certainly the Dicer-KO experiments, which were done by multiple groups. The description toward importance of miRNA in this current version is very superficial. And similar issues can be found across the manuscript. It is highly recommended that the authors should make the efforts to polish the manuscript and make sure the manuscript can communicate with the scientific community with clear and scientific language. .

Reviewer 2 ·

Basic reporting

no comment

Experimental design

no comment

Validity of the findings

no comment

Additional comments

no comment

---

## Round 0.3 · accepted · Accept

The manuscript has been significantly improved.

Thank you again for your submission to PeerJ.